# Convolution on Your 12× Wide Feature: A ConvNet with Nested Design

## Abstract

Transformer stands as the prefered architecture for handling multimodal data under resource-abundant conditions. On the other hand, in scenarios involving resource-constrained unimodal vision tasks, Convolutional Neural Networks (ConvNets), especially smaller-scale ones, can offer a hardware-friendly solution due to the highly-optimized acceleration and deployment schemes tailored for convolution operators. Modern de-facto ConvNets take a ViT-style block-level design, *i.e.* *sequential* design with token mixer and MLP. However, this design choice seems more influenced by the prominence of Transformer in multi-modal domains than by an inherent suitability within ConvNet. In this work, we suggest allocating more proportion of computational resources to spatial convolution layers, and further summarize 3 guidelines to steer such ConvNet design. Specifically, we observe that convolution on 12× wide high dimensional features aids in expanding the receptive field and capturing rich spatial information, and correspondingly devise a ConvNet model with *nested* design, dubbed *ConvNeSt*. ConvNeSt outperforms ConvNeXt in the ImageNet classification, COCO detection and ADE20K segmentation tasks across different model variants, demonstrating the feasibility of revisiting ConvNet block design. As a small-scale student model, ConvNeSt also achieves stronger performance than ConvNeXt through knowledge distillation.

## 1 Introduction

Since the emergence of ViT (Dosovitskiy et al., 2020) in 2020, Transformer (Vaswani et al., 2017) has facilitated the unification of various vision-language tasks, heralding a new phase for multimodal learning. Given its adeptness with text and visual data, it rapidly ascended as a hallmark for multi-modal tasks (Li et al., 2022; 2023; Wang et al., 2023b), especially in resource-abundant conditions. In contrast to ConvNet (LeCun et al., 1989), which has been a shining milestone for decades, the surge in Transformer's popularity seems to be attributed to its employment of a generic multi-head self-attention module that, after addressing textual tasks, also succeeded in meets the visual demands.

From a practical perspective, however, this universal approach is not a one-size-fits-all solution. For example, for unimodal tasks (Redmon et al., 2016; He et al., 2017) (object recognition, detection, segmentation, *etc.*) or scenarios such as building vision encoders in multimodal models (Radford et al., 2021), ConvNets already serve as a cost-effective choice to meet the requirements, especially in resource-constrained settings (Tan & Le, 2019). The reason being, self-attention possesses a quadratic computational complexity, coupled with limited hardware implementation solutions. While there are variants of attention aiming for speedup and reduced resource overheads (Han et al., 2023; Chen et al., 2023), they often compromise the integrity of the original attention function. In contrast, convolution operators benefit from highly optimized acceleration techniques (Mathieu et al., 2014; Winograd, 1980), advanced training strategies (Ding et al., 2021) and versatile compression methods (Han et al., 2016), facilitating their deployment in real-world applications.

Although ConvNets retains its suitability for unimodal tasks, their modern design philosophy has often been "enlightened" by ViT: they focus on the token mixer while directly adopting the ViT's block design, *e.g.*, utilizing large-kernel depth-wise convolutions (Ding et al., 2022), high order convolutions (Rao et al., 2022), or even simple Fast Fourier Transformation (FFT)-based alternatives (Rao et al., 2021; Guibas et al., 2022; Huang et al., 2023b). It's perplexing that the sole reliance on the Transformer seems merely due to its dominance in multimodal tasks. In fact, this design heavily

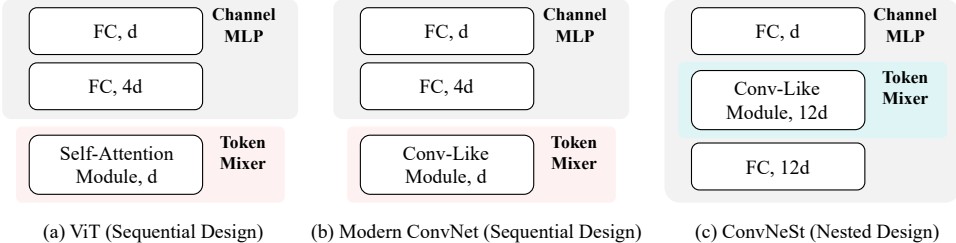

Figure 1: Overall block design of ViT, Modern ConvNet, and ConvNeSt, respectively. Modern ConvNets replace self-attention with enhanced convolution modules as token mixers, but still operating in low dimensions. Our ConvNeSt conducts convolution in high dimensions with $12\times$ wide feature.

allocates computational resources (parameters and FLOPs) to the MLP, rather than spatial convolution. Such ViT-inspired design may offer limited strength in low-resource, unimodal scenarios.

*How should the block structure of ConvNets be designed in such cases?* In this work, we suggest to perform convolution on high dimensional features, aiming for ConvNets to allocate more resources to convolutions over MLP, thereby obtaining meaningful representations. Specifically, we start by placing the convolution between MLPs to utilize its expanded features, albeit it is uncommon in modern ConvNets. Since the number of learnable weights in the convolution layers increases, optimizing these parameters also becomes challenging. Thus, in subsequent sections, we systematically investigate designing ConvNets with such characteristics that ensure both classification accuracy and rich information extraction, facilitating their use in downstream vision tasks.

Through a series of analyses regarding feature distribution, *Effective Receptive Field (ERF)* (Luo et al., 2016), and the loss landscape (Li et al., 2018), we summarize our methodology of effectively convoluting on wide features into 3 guidelines: **1**) pre-activation and post-normalisation refines the feature distribution and facilitates optimization; **2**) large expansion ratio augments the receptive field while maintaining ImageNet classification efficacy; **3**) Although BatchNorm (BN) in ConvNeSt bolsters classification, it exhibits adverse impacts on ERF and the loss landscape.

Based on the above guidelines, we build up a *pure ConvNet with NeSted block design*, dubbed as *ConvNeSt*, that performs convolution on $12 \times$ wide features, as shown in Fig. 2. ConvNeSt, as theoretically demonstrated in Sec. 3.1, allocates increased computational resources to convolutions, offering potential for modeling complex inputs. And the central kernel alignment (CKA) (Kornblith et al., 2019) results in Sec. 3.2 also reveal that the intermediate high dimensional features can capture useful information with considerable separability.

On ImageNet classification (Deng et al., 2009), ConvNeSt consistently outperforms ConvNeXt across all 7 model sizes, 2 structures (isotropic and hierarchical), and 2 optimization paradigm (supervised training and knowledge distillation (Hinton et al., 2015)). Moreover, the advantages of ConvNeSt are more evident when the model becomes smaller, showing ConvNeSt's suitability for resource-constrained scenarios. On COCO (Lin et al., 2014) object detection/segmentation and ADE20K (Zhou et al., 2019) semantic segmentation, ConvNeSt also demonstrates powerful performance and outperform its counterpart. We hope that our exploration will lead to a reconsideration of appropriate application scenarios and targeted architectural designs for ConvNets.

## 2 THE ROADMAP OF CONVNEST

In this section, we introduce the evolution path from ConvNeXt to ConvNeSt. Along the way, we stick to the "convolution on wide features" design and adjust the architecture to fully fullfill this design's potential. To ensure a fair comparison, we adjust the channel width and network depth of the model in all experiments to approximately match the computational complexity of ConvNeXt-T (29M parameters and 4.5G FLOPs). All experiments are trained 300 epochs on ImageNet-1K (Deng et al., 2009). Detailed experimental setup is provided in Sec. A of the Appendix.

### 2.1 CONVOLUTION ON HIGH DIMENSIONAL FEATURES

A simple method to achieve convolution in high dimensional features, with minimal model alterations, is to inject linear transformation layers besides the convolution to increase and reduce the feature

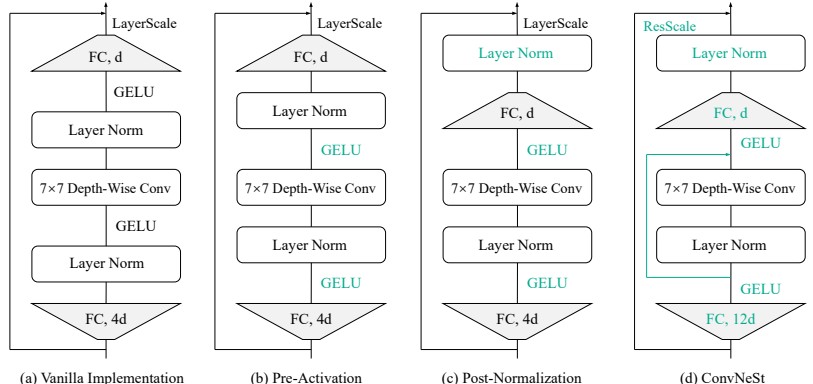

Figure 2: Block design details. (a) Vanilla implementation of convoluting in high dimensional features. (b) We do the normalization after the activation function. (c) The second normalization layer is positioned at the low-dimensional features. (d) A ConvNeSt block.

dimensions, respectively. Thus, we first move the depth-wise convolution in the ConvNeXt-T block to a position between the two linear layers, thereby transitioning the token mixer and MLP from the *sequential* structure to a *nested* one, as shown in Fig. 2(a). Following ViT, two LayerNorm (LN) (Ba et al., 2016) layers are introduced in each block, and we place them between convolutional layers to stabilize the training process. Unlike ViT's single activation function per Block, we added a GELU (Hendrycks & Gimpel, 2016) function after each normalization, following the convention of ConvNets (He et al., 2016; Han et al., 2020). As shown in Tab. 1, such design strategy, which is referred to as "vanilla implementation (VI)", results in an inferior performance of $81.64\%$ top-1 accuracy, compared to the baseline ConvNeXt ($82.1\%$). We argue the reason for the accuracy drop is the optimization difficulty due to such a naive architectural modification.

Thus, we next explored a series of architectural adjustments to unleash the potential of convolution on wide features. Our efforts can be summarized as a roadmap with the following 3 steps: 1) micro design, 2) scaling expansion ratio, 3) normalization layers. After each step, we summarize a guideline for crafting ConvNets with nested design. We adopt the same training recipe for all experiments to ensure that the improvement is solely attributed the architecture adjustment.

Table 1: Results for ConvNeSt-T with different micro designs compared to ConvNeXt. Modification indicates the specific strategy for convolution on high dimensional features.

| Modification | Speci. | Top-1 Acc. |
|---|---|---|
| - | ConvNeXt | **82.1** |
| Vanilla Implementation | Fig. 2 (a) | 81.64 |
| Pre-Activation | Fig. 2 (b) | 81.82 |
| Post-Normalization | Fig. 2 (c) | 81.95 |

Table 2: Results for ConvNeSt-T with different expansion ratios (ER), input channels (C), hidden dimensions (D) and blocks (B). P.: Params (M), F.: FLOPs (G).

| Configuration | P.&F. | Top-1 Acc. |
|---|---|---|
| ER=4, C=96, D=384, B=(3,3,8,3) | 28, 4.5 | 81.95 |
| ER=6, C=80, D=480, B=(3,3,8,3) | 29, 4.7 | **82.03** |
| ER=8, C=72, D=576, B=(3,3,6,3) | 28, 4.6 | 81.89 |
| ER=10, C=64, D=640, B=(3,3,6,3) | 28, 4.6 | 81.75 |
| ER=12, C=56, D=672, B=(3,3,8,3) | 28, 4.8 | 81.99 |
| ER=16, C=48, D=768, B=(3,3,8,3) | 28, 4.8 | 81.66 |

## 2.2 MICRO DESIGN

We fisrt investigate the factors that affect the optimization difficulty of ConvNeSt from an architectural perspective, supported by visualization results for validation.

**Guideline 1: pre-activation and post-normalisation help reshape feature distribution and ease the optimization difficulty.**    Convolutions with significantly more input channels may yield outputs with a different feature distribution. To this end, we visualize the input feature distribution of the final depth-wise convolution layers in the stage 1 and 4 for both ConvNeXt and ConvNeSt. ConvNeXt's feature histograms show a regular symmetrical bell curve, while ConvNeSt's distribution shows an irregular asymmetrical shape with localised convex ridges, as shown in Fig. 3 (a) and (b). Since the learnable weights of convolutions interact directly with the input feature, irregular input distribution may in turn affect the weight distribution, thereby impacting the learning efficiency and ultimately limiting the model capability. We argue that the irregular feature distribution can be attributed to the non-negativity of GELU function outputs, and thus leverage the normalization layer to symmetrize the

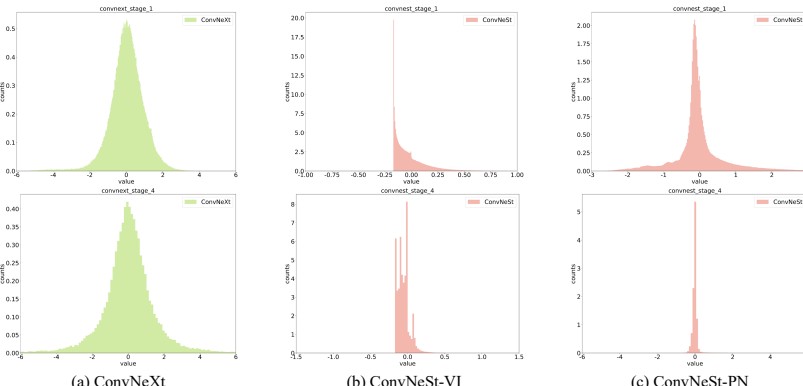

Figure 3: Histograms over the input feature of the final depth-wise convolution layers of stage 1 (up) and stage 4 (bottom) with ConvNeXt (green color) and ConvNeSt (red color).

distribution. Specifically, we placed the two GELU functions before their corresponding LNs, which is referred as "pre-activation (PA)", as shown in Fig. 3 (b). Tab. 1 shows that the PA setting improves the accuracy to $81.82\%$, indicating that a proper distribution is helpful for ConvNeSt optimization.

Additionally, the second LN operates in high-dimensional space. Since LN processes each pixel across channels, increased dimensionality may affect the training stability. We thus apply it to low dimensional features, which is referred to as "post-normalization (PN)", as shown in Fig. 3 (c). With this setting, we get the results of $81.95\%$. From now on, we will use the PN setting in each block.

**Remark 1.** We empirically verifies that the input feature histograms of the PN setting exhibits symmetry, as shown in Fig. 3 (c). Moreover, comparing Fig.5 (a) and (b), the micro design strategies help smooth the loss landscape (Li et al., 2018) of ConvNeSt, thus reducing its optimization difficulty. Detailed feature and weight distribution visualization are shown in Sec. B of the Appendix.

### 2.3 SCALING EXPANSION RATIO

**Guideline 2: large expansion ratio enhances the receptive field without hurting the classification capability on ImageNet.** In this section, we study the ratio of the hidden dimension of the depth-wise convolution to the basic input dimension, *i.e.*, the expansion ratio. Specifically, we vary it by adjusting the ratio between the input and output channel numbers of the linear layer, uniformly spanning from 4 to 16. We also modify the model's width and depth to roughly maintain computational complexity. Note that models with higher expansion ratio have a lower number of basic channels, but their convolution is still performed in a relatively high dimension. We show in Tab. 2 the hyper-parameter settings and hidden dimension for different architectures. Although a large expansion ratio such as 8 or 12 is not popularly adopted in vision backbones, the performance remains potentially powerful. For example, increasing expansion ratio from 4 to 12 still slightly enhances performance.

**Remark 2.** Next, we explore the influence of expansion ratio on *Effective Receptive Field (ERF)* (Luo et al., 2016). The ERF denotes the size of the region within a trained model from which an output at a specific location gathers information about the input. Some work have demonstrated that the ERF can be enlarged by increasing the convolution kernel size (Ding et al., 2022; Liu et al., 2023a), with the expectation of improving the performance of downstream tasks. Different to these works that expand kernel size, we focus on the expansion ratio that affects the hidden dimension of the convolution features, and aim to find the correlation between this and the ERF. Thus, We follow Kim et al. (2021); Ding et al. (2022) to visualize the ERF of ConvNeXt and ConvNeSt with expansion ratios of 6, 12, as illustrated in Fig. 4 (a), (b), and (c). As the expansion ratio increases, ConvNeSt shows a trend to obtain a slightly large ERF. Quantitative analysis about ERF can be found in Sec. C of the Appendix. Based on the consideration between the ERF and accuracy, we use an expansion ratio of 12.

### 2.4 NORMALIZATION LAYERS

**Guideline 3: BatchNorm (BN) in ConvNeSt helps improve classification performance but shows negative effect on ERF and loss landscape.** According to Sec. 2.2, normalization layers reshapes the feature distribution and reduces the optimization difficulty. ConvNeSt employs its two normalization layers at high and low dimensional features respectively. Thus, we substituted the

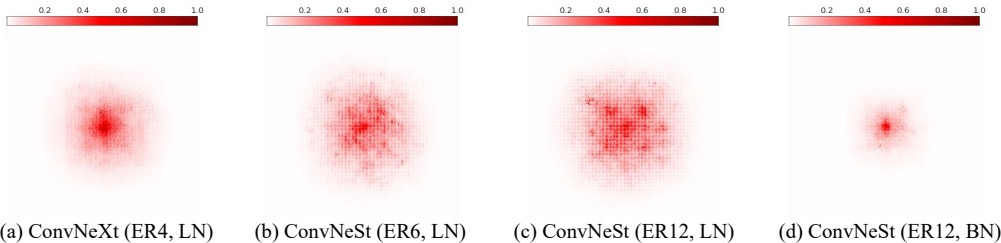

(a) ConvNeXt (ER4, LN)  (b) ConvNeSt (ER6, LN)  (c) ConvNeSt (ER12, LN)  (d) ConvNeSt (ER12, BN)

Figure 4: The *Effective Receptive Field (ERF)* of ConvNeSt with different variants. Augmented expansion ratio (ER) effectively yields a larger ERF, while BN tends to exert an inhibitory effect.

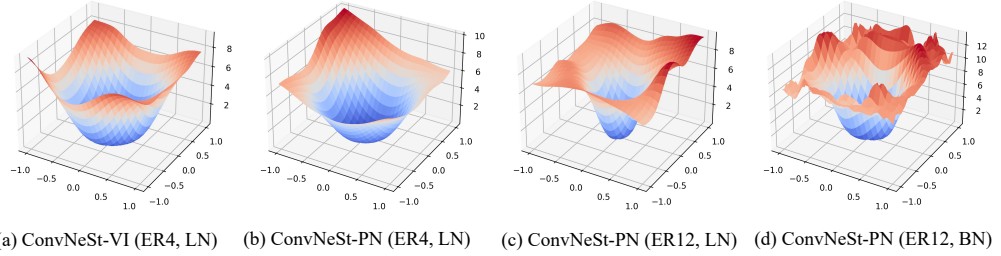

(a) ConvNeSt-VI (ER4, LN) (b) ConvNeSt-PN (ER4, LN) (c) ConvNeSt-PN (ER12, LN) (d) ConvNeSt-PN (ER12, BN)

Figure 5: Optimization landscape different variants of ConvNeSt. ER indicates expansion ratio, and LN, BN refer to using LayerNorm, BatchNorm in the basic block, respectively.

two LNs with BNs in various combinations, resulting in 4 settings, as shown in Tab. 3. We report the ImageNet accuracy and the high-contribution area ratio that positively reflect ERF for different normalization settings, following Ding et al. (2022); Kim et al. (2021). Although BN shows favorable effect for classification accuracy (82.23%), the scope of the high-contribution image pixels has been drastically reduced, suggesting a damage to the ERF, as shown in Tab. 3 and Fig. 4 (d).

**Remark 3.** We hypothesize that the nested design of ConvNeSt may not align well with the functionality of BN. We additionally visualize the loss landscape, as plotted in Fig. 5 (c) and (d). The results indicate that BN renders the loss landscape chaotic and challenging for optimization. Thus, to alleviate potential impacts of a limited ERF on downstream tasks, we use the two LN setting.

Table 3: Test accuracy of ImageNet-1K and quantitative results (high contribution area ratio at different thresholds $t$) of the ERF of the ConvNeSt-T model with the expansion ratio (ER) as 12 using different normalization layers in high dimension (HD) and low dimension (LD): the values positively correlate with ERF.

| Model | Norm (HD) | Norm (LD) | Top-1 Acc. | $t = 20\%$ | $t = 30\%$ | $t = 50\%$ | $t = 99\%$ |
|---|---|---|---|---|---|---|---|
| ConvNeSt-T (ER12) | BN | BN | **82.23** | 0.7 | 1.4 | 3.3 | 44.5 |
| ConvNeSt-T (ER12) | LN | BN | 81.95 | 1.8 | 3.2 | 6.7 | 54.7 |
| ConvNeSt-T (ER12) | BN | LN | 82.10 | 2.4 | 4.0 | 8.1 | 58.5 |
| ConvNeSt-T (ER12) | LN | LN | 81.99 | 3.1 | 5.0 | 9.9 | 68.7 |

## 2.5 OTHER MODIFICATIONS

In this section, we empirically present some useful modifications inspired by modern architecture.

**Scaling branch output.** *LayerScale* (Touvron et al., 2021b) can be viewed as a bias-free affine transformation of the Transformer's residual branch and is adopted by modern visual architectures (Liu et al., 2022; Yu et al., 2022a). It consists of learnable parameters of output dimension size, which are element-wise multiplied with the residual branch output during forward inference and updated with model weights during backpropagation. *ResScale*, employed by Shleifer et al. (2021); Yu et al. (2022b), shifts this affine transformation to the shortcut connection. According to Fig. 1 (c), ConvNeSt's final LN already contains affine transformations that are redundant with the functionality of the *LayerScale*, thus we adopt the *ResScale*, which improves the accuracy to 82.14%.

**Stage compute ratio.** The allocation of computation resource at different stages and is believed to potentially affect the model performance. Following Liu et al. (2021; 2022), we assign more computation to stage3, altering the number of blocks per stage from (3,3,8,3) to (3,3,14,3), and appropriately reducing the model width. The accuracy is further enhanced to 82.44%.

**Dual shortcut.** A Transformer block consists of two sequential residual sub-modules, each equipped with a shortcut branch to facilitate the optimization (He et al., 2016). Inspired by such structure philosophy, we introduce a dual shortcut approach. The two shortcuts are arranged in a nested manner, following the nested block design, as illustrated in Fig. 2. The added shortcut ensures that convolution and LN predominantly learn the residual of intermediate features, improving the accuracy as $82.65\%$.

**Closing remarks.** Here we conclude our exploration journey and introduce ConvNeSt, a high dimensional convolution with a nested design. By elevating the expansion ratio to 12, ConvNeSt achieves a more ideal ERF. With more efficient enhancements equipped, ConvNeSt outperforms ConvNeXt on ImageNet classification. Given our initial intent with general backbone to encompass rich intermediate features, the subsequent sections aim to address: 1) The capacity of ConvNeSt to extract richer information from high dimensional feature. 2) The scalability of this nested approach. 3) Whether the design's ERF preference confers advantages to downstream tasks.

## 3 ANALYSIS OF CONVNEST ARCHITECTURE

### 3.1 THEORETICAL COMPLEXITY ANALYSIS

We provide two additional analyses for a more comprehensive insight into the ConvNeSt architecture. We first showcase, through a detailed theoretical complexity analysis of a single Block, that ConvNeSt increases the allocation of parameters and FLOPs to the token mixer (convolution) as the expansion ratio rises. Given a ConvNeSt Block with expansion ratio $r$ and hidden dimension $d$ defined in Section 2, a depth-wise convolution with kernel size $k \times k$ is used to process the feature with number of channels $rd$, height $h$ and width $w$. Denote the number of parameters and FLOPs for each block as $C_{params}$ and $C_{flops}$, the theoretical ratios of the number of parameters and FLOPs between the token mixer and each block can be expressed as $R_{params} = \frac{2}{1+\sqrt{1+\frac{8C_{params}}{k^4 r}}}$ and $R_{flops} = \frac{2}{1+\sqrt{1+\frac{8C_{flops}}{k^4 hwr}}}$, respectively. The proof is shown in Sec. D of the appendix. Thus, the values of $R_{params}$ and $R_{flops}$ become larger as the expansion ratio increases, meaning that ConvNeSt allocates more computational resources to the convolution and obtains higher dimensional output features. Since the convolutional layer plays a pivotal role in handling spatial interactions and that increased parameters enhance its ability and potential to model complexity, we now move on to empirically analyze these features.

### 3.2 CENTERED KERNEL ALIGNMENT ANALYSIS

We follow Cai et al. (2023) to quantify the implications of the wide features regarding information richness. Specifically, we conduct a Central Kernel Alignment (CKA) (Kornblith et al., 2019; Nguyen et al., 2021) analysis on ConvNeSt-S, assessing the pairwise similarity of output features of the convolutional layer. We first evenly divide these high-dimensional features into 12 groups based on the channel index, and then use the ImageNet-1K validation set to calculate the CKA similarity between each group (Grp for short) and the original input images or their corresponding labels across 4 stages. The similarity matrix is plotted in Fig. 6. As shown, in ConvNeSt-S's early layers (Stage 1 and 2), feature similarity to the original image shows a distinct differences across groups. We surmise that different convolution groups have distinct roles for the image representation: groups with higher CKA similarity (red color) tend to model low-level information, while those with lower similarity (blue color) capture high-level semantic meaning. As the model layers deepen, each group shifts away from texture details. Similar trends appear in ConvNeSt-S's deeper layers (Stage 3 and 4) where feature-label similarity varies notably across groups. The results indicate that different convolution groups may play complementary roles in handling low-level texture and high-level semantic information in images, offering prospects for enhancing the model's representation ability.

## 4 EXPERIMENTS

We build 7 sizes (Woo et al., 2023) of ConvNeSt-A/F/P/N/T/S/B, each of which can be instantiated as a ViT (Dosovitskiy et al., 2020)-style isotropic structure with uniform numbers of channels and spatial tokens throughout and a Swin (Liu et al., 2021)-style hierarchical structure, with an increasing number of channels $C$ and a decreasing number of spatial tokens over $B$ blocks, resulting a total of 14 model variants. The expansion ratio is 12 for all, with hyper-parameters detailed as follows:

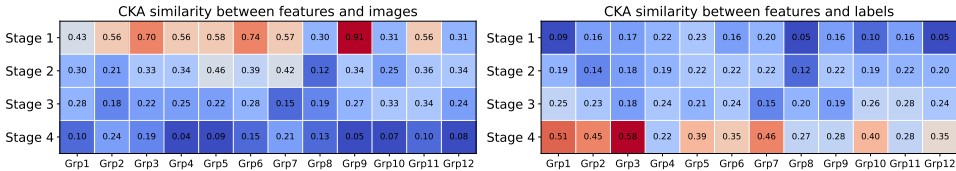

Figure 6: CKA similarities (Kornblith et al., 2019; Nguyen et al., 2021) between 12 groups' output features of depth-wise convolution layers and the input images or labels for 4 stages of ConvNeSt-S.

- ConvNeSt-A *<Hie.>*: $C = (20, 40, 80, 160)$, $B = (2, 2, 9, 2)$ *<Iso.>*: $C = 88$, $B = 13$
- ConvNeSt-F *<Hie.>*: $C = (24, 48, 96, 192)$, $B = (2, 2, 9, 2)$ *<Iso.>*: $C = 92$, $B = 15$
- ConvNeSt-P *<Hie.>*: $C = (32, 64, 128, 256)$, $B = (2, 2, 9, 2)$ *<Iso.>*: $C = 120$, $B = 15$
- ConvNeSt-N *<Hie.>*: $C = (40, 80, 160, 320)$, $B = (2, 2, 12, 2)$ *<Iso.>*: $C = 156$, $B = 15$
- ConvNeSt-T *<Hie.>*: $C = (48, 96, 192, 384)$, $B = (3, 3, 14, 3)$ *<Iso.>*: $C = 192$, $B = 15$
- ConvNeSt-S *<Hie.>*: $C = (56, 112, 224, 448)$, $B = (3, 3, 24, 3)$ *<Iso.>*: $C = 192$, $B = 22$
- ConvNeSt-B *<Hie.>*: $C = (72, 144, 288, 576)$, $B = (3, 3, 28, 3)$ *<Iso.>*: $C = 384$, $B = 22$

## 4.1 IMAGENET CLASSIFICATION

All ConvNeSts are trained for 300 epochs on ImageNet-1K dataset. For ConvNeSt-A/F/P/N with isotropic structure and all other ConvNeSt models, we appropriately adjust the training recipes from Liu et al. (2023b) and Liu et al. (2022), respectively. Noted that hierarchical ConvNeXt-A/F/P/N and isotropic ConvNeXt-A/F/P/N/T accuracies are not reported in the original work (Liu et al., 2022), we thus reimplement the results. Detailed hyper-parameters and training setups are provided in Sec. E of the appendix. Tab. 4 compare our ConvNeSt with modern ConvNets and vision Transformers. ConvNeSt for all model sizes shows strong performance in both isotropic and hierarchical structures compared to modern ConvNets and visual Transformers. Particularly, it outperforms ConvNeXt with on par computational complexity across all model sizes. Additionally, ConvNeSt demonstrates increased strength for smaller models, suggesting that the proposed high-dimensional convolution designs are apt for resource-constrained scenarios of ConvNets.

## 4.2 KNOWLEDGE DISTILLATION

Given our motivation to design architectures apt for resource-constrained scenarios tailored to ConvNet, we also assessed the capability of ConvNeSt in knowledge distillation setting (Hinton et al., 2015) when acting as a small student model. Specifically, we follow Huang et al. (2023a) and use a teacher model SLaK (Liu et al., 2023a) with a large convolutional kernel to assist in the training of student models of the corresponding sizes via NKD (Yang et al., 2022). Detailed hyper-parameters are provided in Sec. E of the appendix. As shown in Tab. 4, ConvNeSt performs better than ConvNeXt for different sizes, demonstrating its suitability to learn effective representations from a teacher model in computation limited applications.

## 4.3 OBJECT DETECTION AND SEGMENTATION

We evaluate ConvNeSt on downstream vision task as a general backbone. On COCO (Lin et al., 2014) object detection/segmentation benchmark, we finetune the Mask R-CNN (He et al., 2017) and Cascade Mask R-CNN (Cai & Vasconcelos, 2018) frameworks using ImageNet pretrained ConvNeSt backbones. All models are optimized using $3\times$ schedule, following Liu et al. (2022).We reimplement all ConvNeXt-A/F/P/N results, as there are no off-the-shelf results from Liu et al. (2022). As shown in Tab. 5, ConvNeSt consistently outperforms ConvNeXt, and ConvNeSt-T outperforms the recent RevCol-T, demonstrating the generalization capability of ConvNeSt as a visual feature extractor.

## 4.4 SEMANTIC SEGMENTATION

We also finetune the UperNet with ImageNet pretrained ConvNeSt backbones on the ADE20K (Zhou et al., 2019) semantic segmentation task. We train all size of models for 160K iterations, following Liu et al. (2022). We also reimplement all ConvNeXt-A/F/P/N results to make a comprehensive

Table 4: **ImageNet-1K classification accuracy** with comparable parameters and FLOPs obtained by ○ Vision Transformers, ● ConvNets that mix spatial tokens on low dimensions, and ● ConvNeSt that works on high dimensions. ↑ represents ImageNet-1K fine-tuning results on high resolution images. ConvNeSt is highlighted.

| Model | Image Size | Params (M) | FLOPs (G) | Top-1 Acc. |
|---|---|---|---|---|
| *ImageNet-1K supervised trained models (isotropic arch.)* | | | | |
| ● ConvNeXt-A (Liu et al.) | $224^2$ | 3.4 | 0.63 | 67.8 |
| ● ConvNeSt-A | $224^2$ | 3.3 | 0.62 | 68.0 |
| ● ConvNeXt-F (Liu et al.) | $224^2$ | 4.0 | 0.75 | 69.7 |
| ● ConvNeSt-F | $224^2$ | 4.1 | 0.77 | 70.2 |
| ○ DeiT-Ti (Touvron et al.) | $224^2$ | 6 | 1.3 | 72.2 |
| ● ConvNeXt-P (Liu et al.) | $224^2$ | 7 | 1.3 | 73.5 |
| ● GFNet-Ti (Rao et al.) | $224^2$ | 7 | 1.3 | 74.6 |
| ● ConvNeSt-P | $224^2$ | 7 | 1.2 | 75.2 |
| ● ConvNeXt-N (Liu et al.) | $224^2$ | 11 | 2.0 | 76.6 |
| ● ConvNeSt-N | $224^2$ | 11 | 2.0 | 78.0 |
| ● ConvNeXt-T (Liu et al.) | $224^2$ | 15 | 2.9 | 78.3 |
| ● GFNet-XS (Rao et al.) | $224^2$ | 16 | 2.9 | 78.6 |
| ● ConvNeSt-T | $224^2$ | 15 | 3.0 | 79.8 |
| ○ DeiT-S (Touvron et al.) | $224^2$ | 22 | 4.6 | 79.8 |
| ● ConvNeXt-S (Liu et al.) | $224^2$ | 22 | 4.3 | 79.7 |
| ● GFNet-S (Rao et al.) | $224^2$ | 25 | 4.5 | 80.0 |
| ● ConvNeSt-S | $224^2$ | 23 | 4.3 | 80.7 |
| ● GFNet-B (Rao et al.) | $224^2$ | 43 | 7.9 | 80.7 |
| ○ DeiT-B (Touvron et al.) | $224^2$ | 86 | 17.6 | 81.8 |
| ● ConvNeXt-B (Liu et al.) | $224^2$ | 87 | 16.9 | 82.0 |
| ● ConvNeSt-B | $224^2$ | 84 | 16.3 | 82.2 |
| *ImageNet-1K supervised trained models (hierarchical arch.)* | | | | |
| ● ConvNeXt-A (Liu et al.) | $224^2$ | 3.7 | 0.55 | 74.1 |
| ● ConvNeSt-A | $224^2$ | 3.6 | 0.67 | 74.4 |
| ● ConvNeXt-F (Liu et al.) | $224^2$ | 5.2 | 0.78 | 75.9 |
| ● ConvNeSt-F | $224^2$ | 5.0 | 0.90 | 76.8 |
| ● PoolFormer-S12 (Yu et al.) | $224^2$ | 12 | 1.8 | 77.2 |
| ● RIFormer-S12 (Wang et al.) | $224^2$ | 12 | 1.8 | 76.9 |
| ● ConvNeXt-P (Liu et al.) | $224^2$ | 9 | 1.4 | 78.6 |
| ● ConvNeSt-P | $224^2$ | 9 | 1.5 | 79.4 |
| ● GFNet-H-Ti (Rao et al.) | $224^2$ | 15 | 2.1 | 80.1 |
| ● ConvNeXt-N (Liu et al.) | $224^2$ | 16 | 2.5 | 80.5 |
| ● ConvNeSt-N | $224^2$ | 15 | 2.7 | 81.0 |

| Model | Image Size | Params (M) | FLOPs (G) | Top-1 Acc. |
|---|---|---|---|---|
| *ImageNet-1K supervised trained models (hierarchical arch.)* | | | | |
| ○ Swin-T (Liu et al.) | $224^2$ | 28 | 4.5 | 81.3 |
| ● GFNet-H-S (Rao et al.) | $224^2$ | 32 | 4.6 | 81.5 |
| ● PoolFormer-S24 (Yu et al.) | $224^2$ | 21 | 3.4 | 80.3 |
| ● RIFormer-S24 (Wang et al.) | $224^2$ | 21 | 3.4 | 80.3 |
| ● PoolFormer-S36 (Yu et al.) | $224^2$ | 31 | 5.0 | 81.4 |
| ● RIFormer-S36 (Wang et al.) | $224^2$ | 31 | 5.0 | 81.3 |
| ● EfficientNet-B4 (Tan & Le) | $380^2$ | 19 | 4.2 | 82.9 |
| ● ConvNeXt-T (Liu et al.) | $224^2$ | 29 | 4.5 | 82.1 |
| ● RevCol-T (Cai et al.) | $224^2$ | 30 | 4.5 | 82.2 |
| ● SLaK-T (Liu et al.) | $224^2$ | 30 | 5.0 | 82.5 |
| ● ConvNeSt-T | $224^2$ | 27 | 4.8 | 82.7 |
| ○ Swin-S (Liu et al.) | $224^2$ | 50 | 8.7 | 83.0 |
| ● GFNet-H-B (Rao et al.) | $224^2$ | 54 | 8.6 | 82.9 |
| ● PoolFormer-M36 (Yu et al.) | $224^2$ | 56 | 8.8 | 82.1 |
| ● RIFormer-M36 (Wang et al.) | $224^2$ | 56 | 8.8 | 82.6 |
| ● PoolFormer-M48 (Yu et al.) | $224^2$ | 73 | 11.6 | 82.5 |
| ● RIFormer-M48 (Wang et al.) | $224^2$ | 73 | 11.6 | 82.8 |
| ● EfficientNet-B5 (Tan & Le) | $456^2$ | 30 | 9.9 | 83.6 |
| ● ConvNeXt-S (Liu et al.) | $224^2$ | 50 | 8.7 | 83.1 |
| ● ConvNeSt-S | $224^2$ | 50 | 9.0 | 83.5 |
| ○ Swin-B (Liu et al.) | $224^2$ | 89 | 15.4 | 83.5 |
| ● RepLKNet-31B (Ding et al.) | $224^2$ | 79 | 15.3 | 83.5 |
| ● EfficientNet-B6 (Tan & Le) | $528^2$ | 43 | 19.0 | 84.0 |
| ● ConvNeXt-B (Liu et al.) | $224^2$ | 89 | 15.4 | 83.8 |
| ● SLaK-B (Liu et al.) | $224^2$ | 95 | 17.1 | 84.0 |
| ● ConvNeSt-B | $224^2$ | 90 | 16.1 | 84.0 |
| ● ConvNeXt-B (Liu et al.) | $384^2$ | 89 | 45.0 | 85.1 |
| ● ConvNeSt-B↑ | $384^2$ | 90 | 47.4 | 85.4 |
| *ImageNet-1K knowledge distilled models (hierarchical arch.)* | | | | |
| ● ConvNeXt-T (Huang et al.) | $224^2$ | 29 | 4.5 | 83.1 |
| ● ConvNeSt-T | $224^2$ | 27 | 4.8 | 83.3 |
| ● ConvNeXt-S (Huang et al.) | $224^2$ | 50 | 8.7 | 84.2 |
| ● ConvNeSt-S | $224^2$ | 50 | 9.0 | 84.4 |

comparison. ConvNeSt also demonstrates advantages for ConvNeXt, RevCol and Swin Transformer on several model scales, showing the effectiveness of nested design on more complex vision tasks.

## 5 RELATED WORK

### 5.1 CONVNETS IN THE POST-VIT ERA

Inspired by ViT, some modern ConvNets work on designing effective token mixers. For example, Rao et al. (2021); Guibas et al. (2022); Huang et al. (2023b); Yu et al. (2022a;b) replace convolution with simple degenerate convolution operations such as FFT-based module, pooling operation, rand mixing, or even identity mapping. Ding et al. (2022); Liu et al. (2023a) expand the convolution kernel to $31 \times 31$ and $51 \times 51$ respectively. Rao et al. (2022) boost the order of the convolution. Our ConvNeSt performs convolution in high dimensions through a nested design, rather than in low dimensions.

### 5.2 INVERTED RESIDUAL BOTTLENECK IN CONVNETS

Inverted residual block is introduced in MobileNetV2 (Sandler et al., 2018) and is further explored in EMO (Zhang et al., 2023). It aligns with our nested design, both of which perform convolution on expanded dimensional features. The differences are: 1) ConvNeSt has a well-designed micro block architecture that reduces the optimization difficulty. 2) ConvNeSt's depth-wise convolutions are done on $12\times$ wide feature, rather than smaller values in MobileNetV2 and EMO. 3) ConvNeSt is a scalable backbone with parameters between 3.7M and 90M, whereas MobileNetV2 (1.4) and EMO-6M, both relatively big models in the original paper, have less than 7M parameters.

Table 5: **Object detection and segmentation results on COCO.** AP results of Swin, X101 and ConvNeXt-T are cited from Liu et al. (2022). FLOPs are measured with a (1280, 800) image input.

| Backbone | FLOPs | $AP^{box}$ | $AP^{box}_{50}$ | $AP^{box}_{75}$ | $AP^{mask}$ | $AP^{mask}_{50}$ | $AP^{mask}_{75}$ |
|---|---|---|---|---|---|---|---|
| *Mask R-CNN 3× schedule* | | | | | | | |
| • ConvNeXt-A (Liu et al.) | 181G | 37.5 | 58.8 | 40.6 | 34.8 | 55.7 | 37.3 |
| • ConvNeSt-A | 183G | 39.1 | 60.8 | 42.2 | 35.9 | 57.5 | 38.1 |
| • ConvNeXt-F (Liu et al.) | 186G | 39.1 | 60.7 | 42.6 | 36.3 | 57.8 | 38.7 |
| • ConvNeSt-F | 188G | 40.8 | 62.6 | 44.2 | 37.3 | 59.6 | 39.8 |
| • ConvNeXt-P (Liu et al.) | 198G | 41.4 | 63.1 | 45.3 | 38.1 | 60.1 | 40.8 |
| • ConvNeSt-P | 200G | 43.1 | 64.8 | 47.2 | 39.0 | 61.5 | 41.8 |
| • ConvNeXt-N (Liu et al.) | 221G | 43.9 | 65.9 | 47.8 | 39.9 | 62.9 | 42.7 |
| • ConvNeSt-N | 224G | 45.1 | 67.0 | 49.4 | 40.6 | 63.9 | 43.8 |
| • ConvNeXt-T (Liu et al.) | 262G | 46.2 | 67.9 | 50.8 | 41.7 | 65.0 | 44.9 |
| ○ Swin-T (Liu et al.) | 267G | 46.0 | 68.1 | 50.3 | 41.6 | 65.1 | 44.9 |
| • ConvNeXt-T (Liu et al.) | 262G | 46.2 | 67.9 | 50.8 | 41.7 | 65.0 | 44.9 |
| • ConvNeSt-T | 268G | 47.3 | 68.7 | 51.6 | 42.1 | 65.9 | 45.2 |
| *Cascade Mask R-CNN 3× schedule* | | | | | | | |
| • ConvNeXt-A (Liu et al.) | 659G | 44.2 | 62.4 | 48.1 | 38.5 | 59.7 | 41.5 |
| • ConvNeSt-A | 661G | 45.3 | 63.8 | 49.4 | 39.3 | 60.7 | 42.2 |
| • ConvNeXt-F (Liu et al.) | 664G | 45.3 | 63.6 | 49.5 | 39.6 | 61.1 | 42.6 |
| • ConvNeSt-F | 666G | 46.5 | 65.0 | 50.6 | 40.3 | 62.4 | 43.5 |
| • ConvNeXt-P (Liu et al.) | 677G | 47.0 | 65.3 | 51.1 | 41.0 | 62.8 | 44.3 |
| • ConvNeSt-P | 679G | 48.1 | 66.9 | 52.2 | 41.7 | 64.0 | 45.1 |
| • ConvNeXt-N (Liu et al.) | 699G | 48.7 | 67.2 | 53.1 | 42.2 | 64.7 | 45.6 |
| • ConvNeSt-N | 703G | 49.3 | 67.9 | 53.4 | 42.7 | 65.8 | 46.0 |
| • X101-64 | 972G | 48.3 | 66.4 | 52.3 | 41.7 | 64.0 | 45.1 |
| ○ Swin-T (Liu et al.) | 745G | 50.4 | 69.2 | 54.7 | 43.7 | 66.6 | 47.3 |
| • ConvNeXt-T (Liu et al.) | 741G | 50.4 | 69.1 | 54.8 | 43.7 | 66.5 | 47.3 |
| • RevCol-T (Cai et al.) | 741G | 50.6 | 68.9 | 54.9 | 43.8 | 66.7 | 47.4 |
| • ConvNeSt-T | 746G | 51.0 | 69.8 | 55.1 | 44.1 | 67.4 | 47.6 |

Table 6: **Semantic segmentation results on ADE20K.** FLOPs are measured with a (2048, 512) image input.

| Backbone | crop size | Params | FLOPs | $mIoU_{ss}$ | $mIoU_{ms}$ |
|---|---|---|---|---|---|
| *UperNet 160K iterations* | | | | | |
| • ConvNeXt-A (Liu et al.) | $512^2$ | 32M | 852G | 37.0 | 37.3 |
| • ConvNeSt-A | $512^2$ | 31M | 852G | 38.6 | 39.4 |
| • ConvNeXt-F (Liu et al.) | $512^2$ | 34M | 857G | 38.9 | 39.6 |
| • ConvNeSt-F | $512^2$ | 33M | 858G | 40.4 | 41.4 |
| • ConvNeXt-P (Liu et al.) | $512^2$ | 39M | 871G | 41.3 | 41.6 |
| • ConvNeSt-P | $512^2$ | 36M | 871G | 42.3 | 43.2 |
| • ConvNeXt-N (Liu et al.) | $512^2$ | 46M | 895G | 43.4 | 44.2 |
| • ConvNeSt-N | $512^2$ | 43M | 896G | 45.1 | 45.9 |
| ○ Swin-T (Liu et al.) | $512^2$ | 60M | 945G | 44.5 | 45.8 |
| • ConvNeXt-T (Liu et al.) | $512^2$ | 60M | 939G | 46.0 | 46.7 |
| • RevCol-T (Cai et al.) | $512^2$ | 60M | 937G | 47.4 | 47.6 |
| • ConvNeSt-T | $512^2$ | 56M | 941G | 47.8 | 48.5 |
| ○ Swin-S (Liu et al.) | $512^2$ | 81M | 1038G | 47.6 | 49.5 |
| • ConvNeXt-S (Liu et al.) | $512^2$ | 82M | 1027G | 48.7 | 49.6 |
| • RevCol-S (Cai et al.) | $512^2$ | 90M | 1031G | 47.9 | 49.0 |
| • ConvNeSt-S | $512^2$ | 79M | 1030G | 48.8 | 49.6 |
| ○ Swin-B (Liu et al.) | $512^2$ | 121M | 1188G | 48.1 | 49.7 |
| • ConvNeXt-B (Liu et al.) | $512^2$ | 122M | 1170G | 49.1 | 49.9 |
| • RevCol-B (Cai et al.) | $512^2$ | 122M | 1169G | 49.0 | 50.1 |
| • ConvNeSt-B | $512^2$ | 120M | 1182G | 49.4 | 50.3 |

## 6 CONCLUSION

Transformer that allows flexible processing of visual and textual information, is widely popular in the multimodal domain. We argue that for resource-constrained unimodal scenarios, ConvNet is still a cost-effective choice due to its simplicity, efficiency, and ease of deployment. Post-ViT era ConvNets have been influenced by ViT, but we believe their block architecture merits re-exploration. We propose a nested design of ConvNet with convolution at 12× wide features, dubbed ConvNeSt. It not only outperforms ConvNeXt on standard vision benchmarks, but also demonstrates the information richness of intermediate features and the efficiency of knowledge distillation.

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

## A    DETAILED HYPER-PARAMETERS OF SEC.2

We provide the experimental setups of the main paper Sec.2, as shown in Tab. 7.

## B    VISUALIZATION RESULTS

We provide more visualization results of feature and weight distribution besides main paper Fig.3. Fig. 7 and Fig. 8 illustrates the histograms of the input feature and trainable weight distribution of the last depth-wise convolution layers in stage 2/3 and stage 1/4 for ConvNeXt, ConvNeSt-VI (Vanilla Implementation setting) and ConvNeSt-PN (Post-Normalization setting), respectively. Feature in ConvNeXt exhibit a regular symmetric distribution. However, under the Vanilla Implementation setting, ConvNeSt, influenced by the non-negativity of GELU (Hendrycks & Gimpel, 2016) activation functions, shows clear unilateral and asymmetry in feature distribution. The Post-Normalization configuration also demonstrates a symmetric feature distribution, consistent with the observation of Fig.3 in the main paper. Additionally, we observed that Post-Normalization leads to a bimodal distribution in the network's deeper weights, while ConvNeXt's weights display a symmetric unimodal distribution.

## C    QUANTITATIVE ANALYSIS OF ERF

We provide the detailed quantitative results about the *Effective Receptive Field (ERF)* (Luo et al., 2016) of ConvNeXt-T (Liu et al., 2022) and ConvNeSt-T models at various expansion ratio settings of Fig.5 in the main paper. Top-1 ImageNet accuracy and the high-contribution area ratio at different thresholds $t$ and different model specifications are reported, following Ding et al. (2022); Kim et al. (2021). The "Specification" column in Tab. 8 corresponds to the ERF visualization results in Fig.5 in the main paper. As shown in Tab. 8, as the expansion ratio increases, the area ratios $r$ of the ConvNeSt model also show growth trends, which suggests a positive effect on ERF. This phenomenon is consistent with the visualization results in Fig.5 in the main paper.

## D    PROOF AND VISUALIZATION OF THEORETICAL COMPLEXITY

Consider a ConvNeSt block as defined in Section 2 of the main paper. Compared to the Fully Connected layer and depth-wise convolution, the scaling factor of the ResScale operation in the ConvNeSt block imposes negligible theoretical complexity. Thus, $C_{params}$ and $C_{flops}$ for each block can be calculated as:

$$
\begin{aligned}
C_{params} &= rd^2 + k^2rd + rd^2, \\
C_{flops} &= hw(rd^2 + k^2rd + rd^2).
\end{aligned}
\tag{1}
$$

where $r, d$ denotes the expansion ratio and hidden dimension of the block, $k \times k$ is the kernel size of the depth-wise convolution, $h, w$ denotes the feature resolution, as defined in Sec.2 of the main paper.

Thus, $R_{params}$ and $R_{flops}$ in the main paper can be expressed as:

$$
R_{params} = \frac{k^2rd}{2rd^2 + k^2rd} = \frac{1}{\frac{2d}{k^2} + 1} = \frac{2}{1 + \sqrt{1 + \frac{8C_{params}}{k^4r}}}
\tag{2}
$$

$$
R_{flops} = \frac{k^2rd}{2rd^2 + k^2rd} = \frac{1}{\frac{2d}{k^2} + 1} = \frac{2}{1 + \sqrt{1 + \frac{8C_{flops}}{k^4hwr}}}
\tag{3}
$$

Then, the $R_{params}$ and $R_{flops}$ in the main paper follow.

In order to observe the trend of $R_{params}$ and $R_{flops}$ as a function of the expansion ratio $r$, we randomly set $k = 3$ and $C_{params} = 10^6$. Thus, the function $R_{params}$ is defined as $f(r)$:

$$f(r) = \frac{2}{1 + \sqrt{1 + \frac{8 \times 10^6}{81r}}} \tag{4}$$

We plot the $f(r)$ in Fig. 9. With the increase of expansion ratio, $f(r)$ also exhibits an upward trend.

## E   DETAILED HYPER-PARAMETERS OF SEC.4

We provide the experimental setups of all hierarchical and isotropic ConvNeSt models and ConvNeXt reimplementation in Sec.4 of the main paper, as shown in Tab. 9 and Tab. 10, respectively.

We further provide the knowledge distillation hyperparameters of ConvNeSt models, as illustrated in Tab. 11.

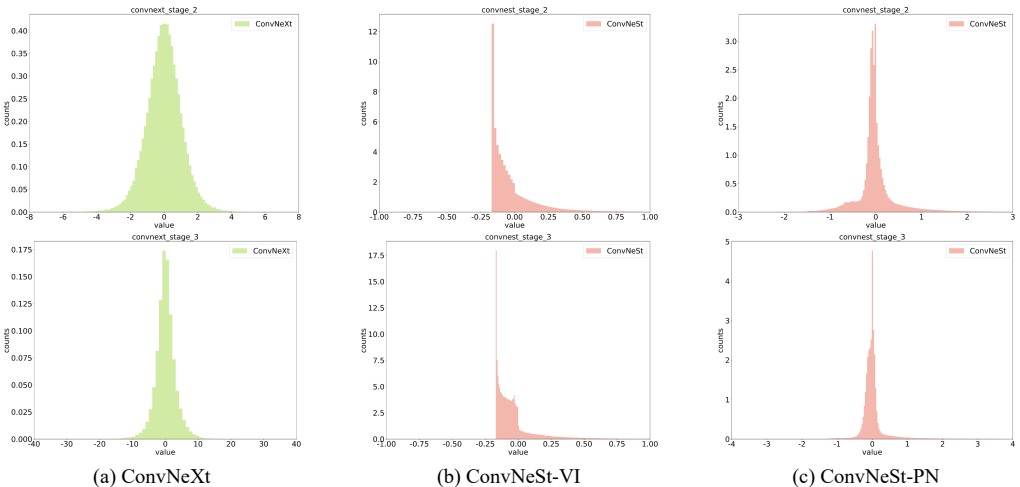

(a) ConvNeXt         (b) ConvNeSt-VI         (c) ConvNeSt-PN

Figure 7: Histograms over the input feature of the final depth-wise convolution layers of stage 2 (up) and stage 3 (bottom) with ConvNeXt (green color) and ConvNeSt (red color).

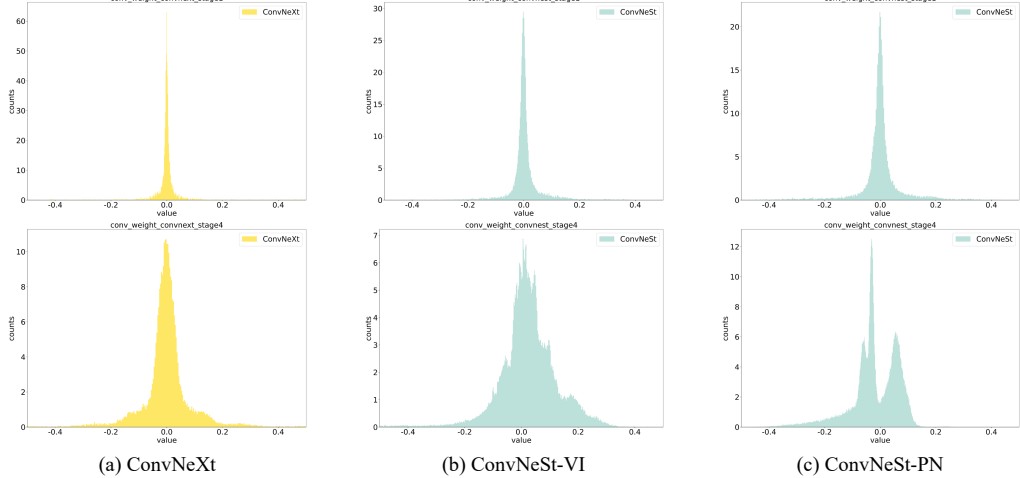

(a) ConvNeXt         (b) ConvNeSt-VI         (c) ConvNeSt-PN

Figure 8: Histograms over the weight of the final depth-wise convolution layers of stage 1 (up) and stage 4 (bottom) with ConvNeXt (yellow color) and ConvNeSt (cyan color).

Table 7: Training recipe for the exploration of ConvNeSt in Sec.2 of the main paper.

| Hyperparameters | ConvNeSt VI/PA/PN settings |
|---|---|
| Input resolution | $224^2$ |
| Warmup epochs | 20 |
| Batch size | 4096 |
| Peak learning rate | 4e-3 |
| Layer-wise learning rate decay (Bao et al., 2022; Clark et al., 2020) | ✗ |
| AdamW (Loshchilov & Hutter, 2019) momentum | (0.9, 0.999) |
| Weight decay | 0.05 |
| Learning rate schedule | cosine |
| Stochastic depth (Huang et al., 2016) | 0.1 |
| EMA (Polyak & Juditsky, 1992) | 0.9999 |
| Gradient clipping | ✗ |
| Label smoothing (Szegedy et al., 2016) $\varepsilon$ | 0.1 |
| RandAugment (Cubuk et al., 2020) | (9, 0.5) |
| Mixup (Zhang et al., 2018) | 0.8 |
| CutMix (Yun et al., 2019) | 1.0 |
| Random erase | 0.25 |

Table 8: Quantitative analysis about the ImageNet-1K results and the *Effective Receptive Field (ERF)* of ConvNeXt and ConvNeSt models at various expansion ratio settings. We report the high contribution area ratios over different model specifications at presupposed thresholds $t$. The values positively correlate with ERF.

| Model | Specification | Top-1 Acc. | $t=20\%$ | $t=30\%$ | $t=50\%$ | $t=99\%$ |
|---|---|---|---|---|---|---|
| ConvNeXt-T (ER4, LN) | Fig.5(a) | 82.1 | 1.6 | 2.9 | 6.5 | 60.3 |
| ConvNeSt-T (ER6, LN) | Fig.5(b) | 82.03 | 2.7 | 4.4 | 9.0 | 67.8 |
| ConvNeSt-T (ER12, LN) | Fig.5(c) | 81.99 | 3.1 | 5.0 | 9.9 | 68.7 |

Table 9: Training recipe for hierarchical ConvNeSt training and ConvNeXt reimplementation.

| Hyperparameters | ConvNeSt A/F/P/N | ConvNeXt reimpl A/F/P/N | ConvNeSt. T/S/B |
|---|---|---|---|
| Input resolution | | $224^2$ | $224^2$ |
| Training epochs | | 300 | 300 |
| Warmup epochs | | 50 | 20 |
| Batch size | | 4096 | 4096 |
| Peak learning rate | | 4e-3 | 4e-3 |
| Learning rate schedule | | cosine | cosine |
| Layer-wise learning rate decay | | ✗ | ✗ |
| AdamW momentum | | (0.9, 0.999) | (0.9, 0.999) |
| Weight decay | | 0.05 | 0.05 |
| Gradient clipping | | ✗ | ✗ |
| Stochastic depth | | 0 | 0.1/0.4/0.4 |
| EMA | | ✗/✗/✗/0.9999 | 0.9999 |
| Label smoothing $\varepsilon$ | | 0.1 | 0.1 |
| RandAugment | | (9, 0.5) | (9, 0.5) |
| Mixup | | 0.8 | 0.8 |
| CutMix | | 1.0 | 1.0 |
| Random erase | | 0.25 | 0.25 |

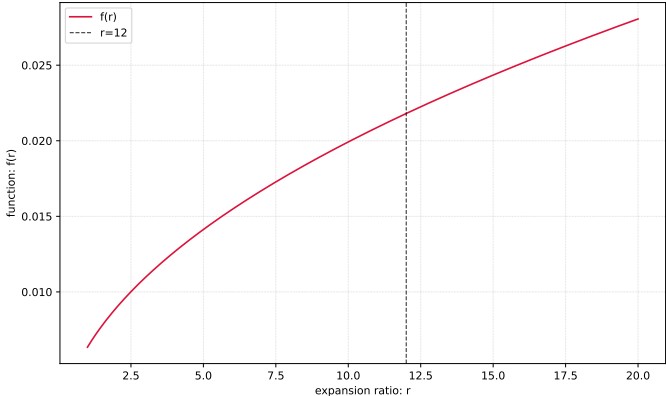

Figure 9: Curve of function $f(r)$. $f(r)$ shows an ascending trend with increasing expansion ratio $r$. ConvNeSt actually takes the value $r = 12$.

Table 10: Training recipe for isotropic ConvNeSt training and ConvNeXt reimplementation.

| Hyperparameters | ConvNeSt | ConvNeXt reimpl | ConvNeSt. |
|---|---|---|---|
| | A/F/P/N/T | A/F/P/N/T | S/B |
| Input resolution | | $224^2$ | $224^2$ |
| Training epochs | | 300 | 300 |
| Warmup epochs | | 50 | 50 |
| Batch size | | 4096 | 4096 |
| Peak learning rate | | 4e-3 | 4e-3 |
| Learning rate schedule | | cosine | cosine |
| Layer-wise learning rate decay | | ✗ | ✗ |
| AdamW momentum | | (0.9, 0.999) | (0.9, 0.999) |
| Weight decay | | 0.05 | 0.05 |
| Gradient clipping | | ✗ | ✗ |
| Stochastic depth | | 0.1 | 0.1/0.4 |
| EMA | | ✗/✗/✗/✗/✗ | ✗/0.9999 |
| Label smoothing $\varepsilon$ | | 0.1 | 0.1 |
| RandAugment | | (9, 0.5) | (9, 0.5) |
| Mixup | | 0.8 | 0.8 |
| CutMix | | 1.0 | 1.0 |
| Random erase | | 0.25 | 0.25 |

## F  VISUALIZATION RESULTS OF CLASS ACTIVATION MAPS

We follow Yu et al. (2022a); Wang et al. (2023a) and provide the Grad-CAM (Selvaraju et al., 2017) results of different pre-trained vision backbones, that is, RSB-ResNet50 (He et al., 2016; Wightman et al., 2021), DeiT-S (Touvron et al., 2021a), ConvNeXt-Small (Liu et al., 2022) and our ConvNeSt-Small. Although the basic components of ConvNeSt and ConvNeXt are similar to each other, they exhibit activation parts with different characteristics: the activation parts of ConvNeXt sometimes exhibit an aggregated distribution similar to that of ConvNet (RSB-ResNet), and sometimes a dispersed distribution similar to that of Transformer (DeiT). On the other hand, ConvNeSt's activation parts do not have any tendency of dispersed distribution, and even show a more concentrated distribution shape than ResNet.

We believe that the reason for this may be that ConvNeSt's nested design of network architecture makes it exhibit different characteristics from ConvNeXt. Specifically, spatial mixing at high dimensional features helps the model to "focus on" concentrated regions when processing images.

Table 11: Knowledge distillation recipe for ConvNeSt in Sec.4 of the main paper.

| Hyperparameters | ConvNeSt |
|---|---|
| | T/S |
| Input resolution | $224^2$ |
| Warmup epochs | 20 |
| Batch size | 2048/4096 |
| Teacher | SLaK-T (Liu et al., 2023a)/SLaK-S |
| Distillation method | NKD (Yang et al., 2022) |
| Peak learning rate | 4e-3 |
| AdamW momentum | (0.9, 0.999) |
| Weight decay | 0.05 |
| Learning rate schedule | cosine |
| Stochastic depth | 0.1 |
| Gradient clipping | ✗ |
| Label smoothing $\varepsilon$ | 0.1 |
| RandAugment | (9, 0.5) |
| Mixup | 0.8 |
| CutMix | 1.0 |
| Random erase | 0.25 |

Note that such a design not only facilitates the improvement of ERF, but also maintains the strong performance of ConvNet in the post-ViT era.

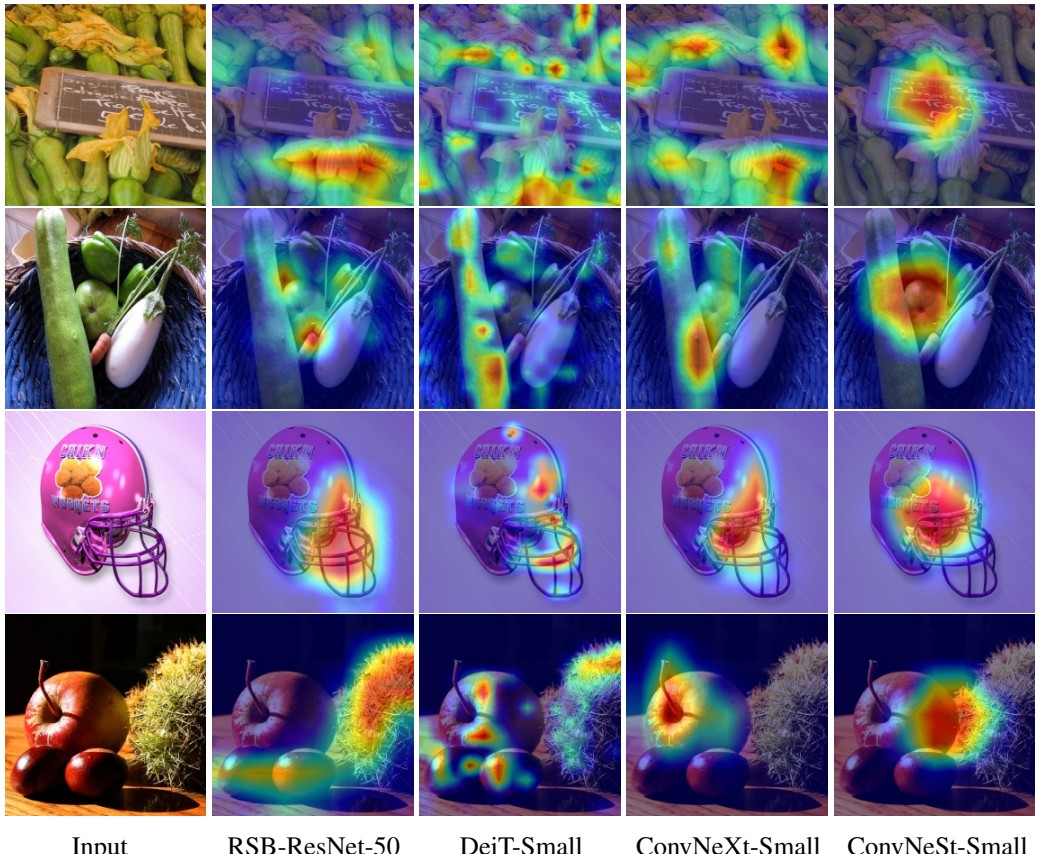

| Input | RSB-ResNet-50 | DeiT-Small | ConvNeXt-Small | ConvNeSt-Small |

Figure 10: Visualization of Class activation maps using Grad-CAM Selvaraju et al. (2017) of different pre-trained vision backbones on ImageNet-1K dataset. The results are plotted by using 4 random images from the validation set.

