# OpenReview forum: "Convolution on Your 12× Wide Feature: A ConvNet with Nested Design"
_ICLR.cc/2024/Conference — ICLR 2024 Conference Withdrawn Submission_

### Official Review · Reviewer_TPf3 · 2023-10-29

**Soundness:** 2 fair
**Presentation:** 3 good
**Contribution:** 2 fair
**Rating:** 3
**Confidence:** 5

**Summary:**

This paper presents a new CNN-based network architecture for visual recognition. Previous works mostly use an expansion ratio of 4 or 6 in the basic building block, which is often an inverted residual one. This paper shows that further increasing the expansion ratio to 12 can result in better performance. Based on this observation, a new network architecture, dubbed ConvNeSt, is proposed. Experiments show that the proposed method performs better than the baselines, including ConvNeXt and RepLKNet.

**Strengths:**

- This paper is well written. The authors have clearly explained the motivation of this paper and it seems that the proposed method is easy to follow.

- A series of ablation experiments are provided to support the arguments by the authors.

- Comprehensive analysis is also provided to help readers better understand the design guidelines of the proposed network.

**Weaknesses:**

- The novelty of this paper is incremental. In the title, the authors have mentioned the importance of wide features in the inverted residual block. However, it seems that the performance improvement is limited as shown in Table 2. This makes me doubt on the effectiveness of the proposed method.

- From Table 2, we can see that further increasing the expansion ratio brings performance drop. To guarantee the model size would not change much, the channel number of the identity path should be shrinked. Have the authors analyzed whether this affects the performance?

- As the channel dimension increases in the middle part of the building block, the learnable convolutional kernels increase consequently. Have the authors analyzed whether this would speed down the inference process? As there are no latency results reported in this paper, it is difficult to infer this?

- In addition, I recommend the authors to report the latency results of the proposed method as done in ConvNeXt. In most cases, FLOPs cannot directly reflect the running speed of a network network model. In many realworld applications, this is very important.

- Most recent classification models, like ConvNeXt, report results based on large-scale models. As I found that when the model size is scaled up to 200M or more, the performance for different models does not change much when a good training recipe is used, for example, the one used in ConvNeXt.

**Questions:**

- The related work section is a little bit thin.
- It seems that the proposed method performs worse than the recent EMO (ICCV'2023) work. Any explanations on this?

---

### Official Review · Reviewer_idR9 · 2023-10-30

**Soundness:** 2 fair
**Presentation:** 3 good
**Contribution:** 2 fair
**Rating:** 5
**Confidence:** 4

**Summary:**

This paper proposes a new ConvNet model, called ConvNeSt, which is designed for resource-constrained unimodal vision tasks.
The authors show a clear roadmap to the block design, such as 1) changes the position of Norm and Activation; 2) modify the expansion ratio etc. Complexity analysis and CKA similarities are also provided. ConvNeSt is validated on the image classification / instance segmentation / semantic segmentation challenges and outperforms state-of-the-art backbones across these tasks.

**Strengths:**

1. The block-level design of ConvNeSt may be helpful to the community.
2. The paper is well-written, and the design roadmap is easy to understand.
3. The visualization and analysis parts are impressive.

**Weaknesses:**

1. The novelty of this paper is limited.
2. The results of tiny/small/base-sized models are given. Will the model design still work on large-sized models?
3. As the authors claim that ConvNets can offer a hardware-friendly solution compared to ViTs, could you show some advantages (like inference speed or memory usage on specific devices) owned by ConvNeSt compared to ViTs. What is more, the comparison with ConvNeXt should also be given.

**Questions:**

Please see the weakness part.

---

### Official Review · Reviewer_N5dm · 2023-11-01

**Soundness:** 3 good
**Presentation:** 3 good
**Contribution:** 2 fair
**Rating:** 6
**Confidence:** 4

**Summary:**

The paper presents ConvNest, a novel convolutional neural network that leverages convolutions across high-dimensional features, significantly enriching the network's feature extraction capabilities. This innovative approach facilitates smoother loss landscapes and enhances learning efficiency, resulting in more distinct and enriched feature representations.

**Strengths:**

The paper shows how ConvNest improve and beat in most cases comparable architectures such as ConvNext in task such as image segmentation and image recognition. Although very slightly in some cases, ConvNest shows a more efficient architecture that reaches similar accuracy but with fewer parameters or resources needed.
The paper  presents different evidence not only accuracy to support the advantage of using ConvNest, for instance an study of the loss landscape and the CKA analysis on the feature space, and even activation maps of selected samples in the apendix. The paper is well written.

**Weaknesses:**

While ConvNest's advancements are clear, I noticed a slight disconnect from the initial discussion on multimodal learning, as the subsequent tasks appeared predominantly unimodal. A refined opening statement could better define the paper's scope, potentially making room for a side-by-side comparison with transformers for a comprehensive analysis.

Although the paper's aim to present a more efficient solution is evident, the impact of relaxing the parameter count constraint on ConvNest's performance remains unclear. Introducing an upper bound model could solidify the argument, illustrating the potential benefits and promising future of adopting the proposed modifications in ConvNest.

**Questions:**

What are the limitations of this model? Why it is unable to reach higher performance?

The activation maps shows great potential for this model, would this model be more robust than other if tested on robust benchmarks?

---

### Official Review · Reviewer_mPKv · 2023-11-02

**Soundness:** 2 fair
**Presentation:** 2 fair
**Contribution:** 2 fair
**Rating:** 3
**Confidence:** 4

**Summary:**

The paper introduces a new design called "ConvNeSt," which is a nested design of ConvNet. This design is proposed to outperform the existing ConvNeXt and other methods on standard vision benchmarks. The paper also touches upon the post-ViT era ConvNets, which have been influenced by the Vision Transformer (ViT) model. The research have tested various models on benchmarks like ImageNet-1K and COCO, comparing their performance in terms of accuracy, FLOPs, and other metrics.

**Strengths:**

The paper introduces a novel nested design for ConvNets, which is expected to enhance their performance on standard vision benchmarks.

Comprehensive testing on well-known benchmarks like ImageNet-1K and COCO provides credibility to the results.

The inclusion of visualizations, tables, and figures likely aids in understanding the results and the model's performance.

**Weaknesses:**

1. As depicted in Fig. 2, the ConvNeSt block offers limited novelty and lacks an in-depth analysis.

2. Increasing the dimension by a factor of 12 will lead to a significant rise in parameters and flops, especially due to the FC layers. Is there any guidance to help us strike a balance between width and computational complexity?

3. When applying convolution to the 12d dimensional feature, even though the flops and parameters remain relatively low in the convolutional layers, there will be a notable reduction in latency. I recommend that the authors report latency measurements on devices, such as GPUs.

4. I suggest revising the figures (e.g., Fig. 2, Fig. 3) in the submission to enhance visualization.

5. The paper omits some crucial baselines, such as VAN [1], SP-VIT [2], ConvFormer[3], MViTv2[4], and others.

[1] Visual Attention Network
[2] SP-ViT: Learning 2D Spatial Priors for Vision Transformers
[3] MetaFormer Baselines for Vision
[4] MViTv2: Improved Multiscale Vision Transformers for Classification and Detection

**Questions:**

Please see above weakness.